# Anisotropic Vector Hysteresis Simulation of Soft Magnetic Composite Materials Based on a Hybrid Algorithm of PSO–Powell

**DOI:** 10.3390/ma13143138

**Published:** 2020-07-14

**Authors:** Xiaojun Zhao, Huawei Xu, Zhenbin Du, Yongjian Li, Lanrong Liu, Zhigang Zhao

**Affiliations:** 1Department of Electrical and Electronic Engineering, North China Electric Power University, Baoding 071003, China; xhw@ncepu.edu.cn; 2Hebei Provincial Key Laboratory of Electromagnetic & Structural Performance of Power Transmission and Transformation Equipment, Baoding 071056, China; sy16@btw.cn (Z.D.); liulanrong@btw.cn (L.L.); 3State Key Lab of Reliability and Intelligence of Electrical Equipment, Hebei University of Technology, Tianjin 300130, China; liyongjian@hebut.edu.cn (Y.L.); iamsam@hebut.edu.cn (Z.Z.)

**Keywords:** SMC material, vector Preisach hysteresis model, rotational magnetization, anisotropy, parameter identification, optimization algorithm

## Abstract

To simulate the anisotropic hysteresis characteristics of soft magnetic composite (SMC) materials accurately, an improved vector hysteresis model was proposed and utilized to adjust the shape of hysteresis curves by introducing two parameters. These two parameters are correlated with the amplitude of the vector Everett function and the projection of magnetic flux density along different directions. An experimental platform was built to measure the two-dimensional (2-D) magnetic properties of the SMC material under rotational magnetizations. The scalar and vector Everett functions were constructed by the measured limiting hysteresis loops. A hybrid optimization strategy based on the particle swarm optimization (PSO) and Powell technique was proposed to identify the parameters of the improved model efficiently and precisely, which significantly improved the local optimization ability of the PSO algorithm. The simulated results strongly agree with the measured ones, and thus the effectiveness of the improved vector model and the parameter identification method proposed in this paper was verified.

## 1. Introduction

Soft magnetic composite (SMC) materials have been extensively used in motors and power electronic devices due to their unique electromagnetic properties such as magnetic and thermal isotropy, diversified processing shapes, and low eddy current loss at medium and higher frequencies [1]. As a kind of complex characteristic inherent in SMC materials, hysteresis has a great impact on the optimal design and analysis of electrical equipment, so accurate measurement and simulation of the hysteresis properties of these materials is critical to their research and application [2].

Traditional scalar simulation of the magnetic performance enables precise results under alternating magnetic excitations. However, the electrical devices usually operate under alternating and rotational excitations in practice, and the magnetic flux density ***B*** and magnetic field strength ***H*** do not always align in the same direction. The iron loss caused by the rotational excitations is greater than that induced by the unidirectional alternating fields [3,4,5]. Therefore, it is of great importance to measure and model the vector hysteresis properties of SMC materials under rotational excitations [6].

Different vector hysteresis models are presented in previous studies to obtain the rotational magnetic characteristics by using some newly developed measuring methods [7]. Stoner and Wohlfarth proposed a vector hysteresis model which was designed as an ensemble of single-domain and uniaxial particles. Nevertheless, it was impossible to fit the asymmetric hysteresis loops and accomplish the parameter identification completely [8]. In view of the defects of the model mentioned above, Mayergoyz presented a classical vector Preisach model to simulate the isotropic vector hysteresis characteristics of magnetic materials [9]. However, the anisotropy magnetic properties under rotational excitations are found even at a relatively low frequency [10,11], so the classical model is no longer applicable in most cases, leading to imperative demands for further improvement. Miklós Kuczmann proposed an improved vector Preisach model to simulate the slight anisotropic behavior of isotropic materials, whereas it indicated great errors at low amplitudes of the magnetic flux density [12]. In addition, M. Enokizono and N. Soda presented the E&S vector hysteresis model considering the anisotropic properties under rotational magnetization. It requires large amounts of experimental data to obtain numerous parameters, which severely limits its application [13].

In this paper, an experimental platform for the measurement of vector magnetic properties was employed to obtain the 2-D magnetic properties of the SMC material under rotational excitations [14,15,16]. By introducing the correlation parameters, the classic vector hysteresis model was modified to simulate the vector hysteresis curves of the SMC material, considering the anisotropy property. Based on the measured limiting hysteresis loops in two orthogonal directions, the first-order reversal curves (FORCs) were numerically generated to construct the scalar and vector Everett functions. A hybrid optimization strategy combining the particle swarm optimization (PSO) algorithm and the Powell technique was proposed for parameter identification in the improved model with good efficiency and precision, which significantly improved the local optimization ability of the PSO algorithm. The validity of the proposed method was proved by comparing the measured results with the simulated ones.

## 2. Material and Methods

### 2.1. Material

The SMC material, SOMALOY^TM^ 500 (developed by Höganäs in Höganäs, Sweden), was employed, which has a maximum magnetic flux density of 2.1 T at a magnetic field strength of 100 kA/m, residual magnetization of 0.25 T, a coercive magnetic field strength of 250 A/m, conductivity of 30 μS/m, a coefficient of thermal conductivity of 17 W/(m∙K) and an initial relative magnetic permeability of 130.

### 2.2. Vector Hysteresis Measurement of SMC Material

The vector magnetic property testing system consisted of a main measuring device, a power amplifier, a magnetic sensing system and a signal-processing unit. The main measuring devices are shown in Figure 1, including the “C-type” core-poles and core-yokes, excitation windings, and three ***B***-***H*** composite sensing coils.

The measuring system was used to get the 2-D magnetic properties of the material. During the experiment, a cubic specimen made of SMC with a size of 22 × 22 × 22 mm^3^ was fixed in a sensing box mounted with ***B***-***H*** sensing coils on six sides to measure the corresponding ***B*** and ***H***.

Figure 2 is the structure diagram of the measuring system. The signal generating system was used as the computing core to provide the analog excitation signals for three high power amplifiers. Then the excitation voltage signals were selected by the impedance matching loop and the resonant circuit to obtain a large excitation current, which acted as the input of the main excitation magnetic circuit so as to generate a rotational magnetic field inside the measured specimen. The sensing coils on the specimen can pick up weak electrical signals, and then these signals are input into the signal processing circuit and collected by a LabView embedded control system. The closed-loop feedback of the system is realized in such way.

In order to ensure the circular trajectory of the 2-D magnetic flux density ***B***, a circular rotational excitation at a low frequency (*f* = 5 Hz) was set on the *x*-*o*-*y* plane. The locus of the vector ***B*** and ***H*** with the increase of excitation current are demonstrated in Figure 3, and the locus of ***H*** exhibits a special feature in an asymmetric and irregular shape due to the slight anisotropic property.

The measured concentric hysteresis loops along two orthogonal directions, i.e., *x* direction and *y* direction, are shown in Figure 4. The hysteresis curves at the saturated magnetic flux density are the limiting hysteresis loops required for identification of the Preisach model.

### 2.3. Hysteresis Modeling Based on Improved Preisach Model

#### 2.3.1. Improved Vector Preisach Model Based on Classical Model

The inverse form of the classical vector Preisach model, which predicts the magnetic field strength ***H*** from the magnetic flux density ***B***, is expressed as follows [9],
(1)$$H\left(t\right)=\int_{-\pi /2}^{\pi /2}e_{\varphi }\left(H_{\varphi }\left(B_{\varphi }\right)\right)d\varphi$$
where *H_φ_*(*B_φ_*) is the scalar magnetic field strength of *H*(*t*) in the direction ***e****_φ_*.

In the numerical calculation, the angle φ ϵ [−π/2, π/2] is evenly divided into *n* cells:(2)$$\varphi_{i}=-\frac{\pi }{2}+\frac{i-1}{n}\pi$$
where *i* = 1,…,*n* and *n* is the number of directions.

The vector magnetic field strength *H*(*t*) in Equation (1) can be expressed as the sum of the scalar magnetic field strength *H_φ_**_i_*(*B_φ_**_i_*) along all the *n* directions, described as follows,
(3)$$H\left(t\right)=\sum_{i=1}^{n}e_{\varphi_{i}}\left(H_{\varphi_{i}}\left(B_{\varphi_{i}}\right)\right)\Delta \varphi$$
(4)$$H_{\varphi_{i}}\left(B_{\varphi_{i}}\right)=\iint_{\alpha \geq \beta }v\left(\alpha ,\beta \right)\gamma_{\alpha \beta }B_{\varphi_{i}}d\alpha d\beta$$
where ∆*φ*
*=* π/*n*. *γ_αβ_* is the simplest hysteresis operator with *α* and *β* corresponding to “up” and “down” switching values of the input, respectively. *v* is the vector Preisach distribution function.

The magnetic flux density in the direction *φ_i_* can be calculated from the two components of the magnetic flux density:(5)$$B_{\varphi_{i}}=B_{x}\cos\varphi_{i}+B_{y}\sin\varphi_{i}$$

As shown in Equation (4), the key to identifying the vector model is to obtain the vector distribution function *v*. There are two main difficulties in identifying *v* directly. On one hand, it is time-consuming to evaluate the double integral in Equation (4) numerically. On the other hand, the determination of the distribution function *v* requires differentiations of experimentally obtained data, which may amplify the inherent experimental errors significantly [9].

In order to eliminate the double integral of *α* and *β*, the discrete Everett function *E* [9] is commonly used to identify the vector hysteresis model numerically. The specific expression of *E* is as follows,
(6)$$E\left(\alpha ,\beta \right)=\iint_{T\left(\alpha ,\beta \right)}v\left(\alpha ,\beta \right)d\alpha d\beta$$
where T(*α*,*β*) is the limit triangle surrounded by *α* and *β* in the Preisach plane.

According to Equation (6), *H_φ_**_i_*(*B_φ_**_i_*) can be expressed by *E* in the following equation:(7)$$H_{\varphi_{i}}\left(B_{\varphi_{i}}\right)=-E\left(B_{0},b_{0}\right)+2\sum_{k=1}^{n_{\varphi_{i}}\left(t\right)}\left[E\left(B_{\varphi_{i},k},b_{\varphi_{i},k-1}\right)-E\left(B_{\varphi_{i},k},b_{\varphi_{i},k}\right)\right]$$
where *B*_0_ and *b*_0_ are the positive and negative saturation values of ***B***. *B_φ_**_i,k_* and *b_φ_**_i,k_*_−1_ represent a series of maximum and minimum values of *B_φ_**_i_*. *n_φ_**_i_*(*t*) is the number of reversal points of the first-order reversal curves (FORCs) [17,18,19] in the direction *φ_i_*.

The function *E* is related to the scalar Everett function *F*, as described in the following equation:(8)$$F\left(\alpha ,\beta \right)=\int_{-\pi /2}^{\pi /2}\cos\varphi E\left(\alpha \cos\varphi ,\beta \cos\varphi \right)d\varphi$$
where function *F* can be generated from the measured data by numerical methods.

The classical vector Preisach hysteresis model given above implies that for a uniformly and rotationally applied ***B***, the locus of ***H*** is circular [20], as shown in Figure 3a. However, the locus of ***H*** exhibits a bit petal-like shape because of the slight magnetic anisotropy property of the SMC material itself, as shown in Figure 3b. It is obvious that the classical vector model is no longer applicable. In order to circumvent the limitations of the classical model above, an improved Preisach model was proposed by introducing two parameters *w* and *z*. These two parameters are connected to the amplitude of Everett function *E* and the projection of magnetic flux density along different directions.
(9)$$H\left(t\right)=\int_{-\pi /2}^{\pi /2}e_{\varphi }\left(H_{\varphi }\left(B_{\varphi }\right)\right)d\varphi ≅\sum_{i=1}^{n}e_{\varphi_{i}}\left(H_{\varphi_{i}}\left(B_{\varphi_{i}}\right)\right)\Delta \varphi$$
(10)$$B_{\varphi_{i}}=B_{x}\mathrm{sign}\left(\cos\varphi_{i}\right){\left|\cos\varphi_{i}\right|}^{w}+B_{y}\mathrm{sign}\left(\sin\varphi_{i}\right){\left|\sin\varphi_{i}\right|}^{w}$$
(11)$$F\left(\alpha ,\beta \right)=\int_{-\pi /2}^{\pi /2}\cos\varphi E\left(\alpha \cos^{w}\varphi ,\beta \cos^{w}\varphi \right)z\left(B_{m}\right)d\varphi$$
where *B*_m_ is the magnitude of ***B***.

From Equations (10) and (11), it can be seen that the parameter *w* generalizes the projection of ***B*** and controls the projection of ***H*** along each discrete direction. It enables the slight anisotropic property of the model. The parameter *z* was used to adjust the function *E* and the amplitude of ***H*** under different *B*_m_ to get more accurate fitted results. When 0 < *w* < 1, the improved model can simulate slight anisotropic hysteresis characteristics of the material, thus making the locus of ***H*** an elliptical or a petal-like shape. In particular, it is in accordance with the classic vector model as *w* = 1.

The two orthogonal components of the magnetic field strength are given by
(12)$$H_{x}=\sum_{i=1}^{n}H_{\varphi_{i}}\cos\varphi_{i}$$
(13)$$H_{y}=\sum_{i=1}^{n}H_{\varphi_{i}}\sin\varphi_{i}$$

#### 2.3.2. Identification Procedure of Improved Vector Preisach Model

According to Equation (11), the numerical identification of scalar Everett function *F* [21] can be represented by function *E* as follows,
(14)$$F\left(\alpha ,\beta \right)=\sum_{i=1}^{n}\cos\varphi_{i}E\left(\alpha \cos^{w}\varphi_{i},\beta \cos^{w}\varphi_{i}\right)z\left(B_{m}\right)\Delta \varphi$$

The function *F* in Equation (14) can be constructed numerically based on the measured limiting hysteresis loops in Figure 4. Their descending branches are used for the generation of the FORCs by the numerical method proposed by Dlala [22], as shown in Figure 5.

According to the identification procedure presented in [9], the scalar Everett functions can be obtained from the interpolation based on FORCs,
(15)$$F\left({\bar{b}}_{u},{\bar{b}}_{v}\right)=\frac{1}{2}\left(H_{\mathrm{forc}}\left({\bar{b}}_{u},{\bar{b}}_{v}\right)-H_{\mathrm{forc}}\left({\bar{b}}_{u}\right)\right)$$
where ${\bar{b}}_{u}$ and ${\bar{b}}_{v}$ respectively correspond to the discrete increasing and decreasing values of *B* on the FORCs. *H*_forc_(${\bar{b}}_{u}$,${\bar{b}}_{v}$) refers to the value of *H* on the FORCs, while *H*_forc_(${\bar{b}}_{u}$) represents the value of ***H*** at the reversal points.

According to Equation (15), the corresponding scalar Everett functions *F_x_* and *F_y_* are depicted in Figure 6, respectively.

The scalar Everett function in the direction *φ* is expressed as the elliptic interpolation of *F_x_* and *F_y_* to approximate the smooth angular behavior,
(16)$$F\left(\alpha ,\beta ,\varphi \right)=\sqrt{F_{x}{}^{2}\left(\alpha ,\beta \right)\cos^{2}\varphi +F_{y}{}^{2}\left(\alpha ,\beta \right)\sin^{2}\varphi }$$

Therefore, the scalar Everett function *F* under the circular rotational magnetization is represented as follows [21],
(17)$$F\left(\alpha ,\beta \right)=\frac{\Delta \varphi }{\pi }\left(F\left(\alpha ,\beta ,0\right)+F\left(\alpha ,\beta ,\frac{\pi }{2}\right)+2\sum_{j=1}^{n-1}F\left(\alpha ,\beta ,\varphi_{j}\right)\right)$$

The generated scalar Everett function *F* is depicted in Figure 7a.

As shown in Equation (14), it should be noticed that the function *E* can be obtained from the scaler Everett function *F* at specific values of *B*_m_, *w*, and *z*. Assuming that *B*_m_ = 1.398 T, *w* = 0.7 and *z* = 2, the corresponding vector Everett function is shown in Figure 7b.

#### 2.3.3. Parameter Extraction of Improved Model Based on Hybrid Optimization Algorithm

The improved model can simulate the anisotropy property of SMC materials by introducing parameters *w* and *z*. The extraction of these two parameters is essential to the simulation accuracy. For full utilization of the strong global search ability of the random optimization algorithm as well as the fast local convergence of the deterministic optimization algorithm, a hybrid optimization strategy combining PSO with the Powell algorithm was presented to implement the parameter extraction.

The mean absolute percent error (MAPE) was used to evaluate the simulation accuracy of the improved vector hysteresis model. Thereafter, the parameter extraction of the improved model can be performed by searching for the minimum value of the objective function *f* given in Equation (18).
(18)$$\min f=\frac{1}{N}\sum_{i=1}^{N}\left|\frac{H_{\mathrm{cal}}\left(i\right)-H_{\mathrm{mea}}\left(i\right)}{H_{\mathrm{mea}}\left(i\right)}\right|\times 100\%$$
where *H*_cal_ and *H*_mea_ refer to the calculated and measured magnetic field strength, respectively. *N* is the total amount of data.

During the initial iteration of the hybrid algorithm, the PSO algorithm was first used to perform a wide-ranging optimization to quickly lock the solution region. Specific steps are as follows.(1)Set the parameters of the PSO algorithm. The particle number *N*, the acceleration factors *c*_1_ and *c*_2_, the inertia factor *ω*, the maximum number of iterations *T* and the initial iteration number *k* are set to 5, 0.3, 0.3, 1, 100 and 1 respectively.(2)Initialize the population. The position *X*(*w_i_*^0^, *z_i_*^0^) of each initial particle *i* is generated randomly within *wi*0 ϵ [0.5,1], *zi*^0^ ϵ [1,1.5]; the range of the initial particles’ velocity *V_i_*^0^ ϵ [*V*_min_, *V*_max_] is set to [0,0.3].(3)Calculate the objective function *f_i_*^0^ of each initial particle to obtain the historical optimal position *P_i_*^0^ = *X_i_*^0^ and the global optimal position *P*_g_^0^ = *P*(*w*_g_^0^, *z*_g_^0^).(4)Update the particle velocity *V_i_^k^* and position *X_i_^k^* by *P_i_^k^*^−1^ and *P*_g_*^k^*^−1^
(19)$$\left\{ \begin{array}{l} V_{i}^{k}=\omega V_{i}^{k-1}+c_{1}r_{1}\left(P_{i}^{k-1}-X_{i}^{k-1}\right)+c_{2}r_{2}\left(P_{g}^{k-1}-X_{i}^{k-1}\right)\\ X_{i}^{k}=X_{i}^{k-1}+V_{i}^{k} \end{array}\right.$$
where *r*_1_ and *r*_2_ are generated randomly within the interval of [0,1].(5)Evaluate the *f_i_^k^* of each particle and update the historical optimal position *P_i_^k^* of each particle as well as global optimal objective position *P*_g_*^k^* as follows
(20)$$\left\{ \begin{array}{l} f_{i}^{k}=f\left(P_{i}^{k}\right)\\ f_{g}^{k}=f\left(P_{g}^{k}\right)=\underset{i=1,\cdots ,N}{\min}\left\{f_{i}^{k}\right\}=\underset{i=1,\cdots ,N}{\min}\left\{f\left(P_{i}^{k}\right)\right\} \end{array}\right.$$
where *f*_g_*^k^* is the global optimal objective function at the *k*^th^ iteration.(6)Determine whether the switching criteria as Inequation (21) is satisfied. If satisfied, the current optimal solution *P*_g_*^k^*= *P*(*w*_g_*^k^*, *z*_g_*^k^*), and the corresponding objective function *f*_g_*^k^,* are transferred to the Powell algorithm and the calculation process is ended. Otherwise, set *k* = *k* + 1 and repeat from step (3).
(21)$$\frac{\sum_{n=1}^{n_{0}}\left|f_{g}^{k+1-n}-f_{g}^{k-n}\right|}{f_{g}^{k}}<\varepsilon$$
where ε = 0.01. *n*_0_ is the number of consecutive iterations, and was set to 10 in this paper.

After receiving the optimal solution provided by the PSO, the Powell algorithm which can converge to the optimal solution efficiently was utilized to optimize the parameters of the vector Preisach model. The specific process is as follows:(1)Initial the basic point of the Powell algorithm: *x*_0_^(1)^ = *x*^(0)^ = *x*(*w*^(0)^, *z*^(0)^) = *P*(*w*_g_*^k^*, *z*_g_*^k^*).(2)Set the parameters of the Powell algorithm: the iteration accuracy *e*, the initial direction ***S***_1_^(1)^, ***S***_2_^(1)^, and the initial iteration number *t,* are set to 0.001, (1,0), (0,1) and 1, respectively.(3)Basic search: start from *x*_0_^(*t*)^ and do a 1-D search along ***S***_1_^(*t*)^ and ***S***_2_^(*t*)^ to obtain the extreme points *x*_1_^(*t*)^ and *x*_2_^(*t*)^ for *f*.(4)Accelerated search: start from *x*_0_^(*t*)^, perform a 1-D search along the conjugate direction ***S***^(t)^ = *x*_2_^(*t*)^ - *x*_0_^(*t*)^ to get the extreme point *x*_3_^(*t*)^.(5)Determine whether the termination condition as Inequality (22) is met. If satisfied, the current optimal solution *x*^*^ = *x*_3_^(*t*)^ = *x*(*w*_3_^(*t*)^, *z*_3_^(*t*)^), and the corresponding optimal value *f*(*x*^*^), are obtained. Otherwise, go to step (6).
(22)$$\left\|x_{3}^{\left(t\right)}-x_{0}^{\left(t\right)}\right\|\leq e$$(6)Calculate the maximum drop $\Delta_{m}^{\left(t\right)}$ of *f* and the corresponding direction ***S***_m_^(*t*)^ as follows.
(23)$$\left\{ \begin{array}{l} \Delta_{m}^{\left(t\right)}=\underset{i=1,2}{\max}\left\{\Delta_{i}^{\left(t\right)}\right\}=\underset{i=1,2}{\max}\left\{f\left(x_{i-1}^{\left(t\right)}\right)-f\left(x_{i}^{\left(t\right)}\right)\right\}\\ S_{m}^{\left(t\right)}=x_{m}^{\left(t\right)}-x_{m-1}^{\left(t\right)} \end{array}\right.$$Calculate the mapping point *x*_map_^(*t*)^ = 2*x*_2_^(*t*)^ − *x*_0_^(*t*)^ along direction ***S***^(*t*)^ and set *f*_1_ = *f*(*x*_0_^(*t*)^), *f*_2_ = *f*(*x*_2_^(*t*)^), *f*_3_ = *f*(*x*_map_^(*t*)^). Update the initial point *x*_0_^(*t* + 1)^ = *x*_3_^(*t*)^ and the search direction ***S***_m_^(*t*)^ = ***S***^(*t*)^ if the Powell condition is satisfied as Inequality (24), and then repeat from step (3). Otherwise, go to step (7).
(24)$$\left\{ \begin{array}{l} f_{3}<f_{1}\\ \left(f_{1}-2f_{2}+f_{3}\right){\left(f_{1}-f_{2}-\Delta_{m}^{\left(t\right)}\right)}^{2}<0.5\Delta_{m}^{\left(t\right)}{\left(f_{1}-f_{3}\right)}^{2} \end{array}\right.$$(7)Update the initial point: *x*_0_^(*t* + 1)^ = *x*_2_^(*t*)^ if $f_{2}<f_{3}$. Otherwise, update the initial point: *x*_0_^(*t* + 1)^ = *x*_map_^(*t*)^. Set *t* = *t* + 1 and repeat from step (3).

The specific calculation flowchart of the parameter extraction based on hybrid algorithm of PSO–Powell is demonstrated in Figure 8.

By using the PSO–Powell hybrid algorithm, the numerical simulation of the anisotropic vector hysteresis properties under circular rotational magnetization was completed by the improved vector hysteresis model, and the detailed procedures are given in Figure 9.

## 3. Results and Discussion

The PSO algorithm and the hybrid optimization strategy of PSO–Powell, respectively, give the same initial values of *w* and *z* to optimize the two parameters, so we can compare the convergence performance of the two algorithms intuitively. The values of the maximum iteration number are set appropriately for the two algorithms, and the termination criteria of the hybrid algorithm is determined by the convergence condition of Powell method.

Based on the PSO algorithm, the variation of MAPE with the number of iterations at *B*_m_ = 1.398 T is demonstrated in Figure 10, which shows strong global optimization ability but worse local search ability. Although it takes only four iterations to reduce MAPE from 21.3698% to 7.3943% in the beginning, the solutions are trapped in local optimum when the values of MAPE vary little. The aforementioned defect indicates that the convergence performance of the PSO algorithm tends to deteriorate when approaching the solution region, which makes it difficult to extract the optimal parameters accurately.

Table 1 lists the parameters *w* and *z*, corresponding values of MAPE, as well as the bounds of relative error extracted by PSO algorithm at *B*_m_ = 0.178 T, 0.423 T, 0.704 T, 1.053 T and 1.398 T. As shown in Figure 11, the comparison between the measured hysteresis curves of vector ***H*** and the simulated results obtained from the extracted parameters were made, which showed considerable discrepancy. This brings a significant obstacle to the magnetic hysteresis property simulation of the SMC material.

In contrast to the convergence performance of the PSO algorithm in Figure 10, Figure 12 as below shows the corresponding variation of MAPE with the iteration steps based on the PSO-Powell hybrid algorithm.

The trend of MAPE variation shows that the PSO algorithm meets the switching criteria automatically when it reaches the 12^th^ iteration, thereafter the Powell algorithm is started with the previously optimized results as the initial solution. Then it takes only four iterations to achieve the convergence. These results show that the hybrid algorithm of PSO–Powell exhibits a faster convergence speed than the PSO algorithm.

In contrast to the values in Table 1, the values of the parameters and corresponding error extracted by the hybrid algorithm are listed in Table 2. The values of MAPE and bounds of relative error are, respectively, adjusted below 5.9571% and 10.7819%, which shows a significant reduction in the error of the improved model and a higher accuracy than that of the PSO algorithm.

Based on the extracted parameters in Table 2, the comparison between the measured hysteresis curves of vector ***H*** and simulated results optimized by the hybrid algorithm was made and is depicted in Figure 13. The simulated results strongly agree with the measured ones when *B*_m_ varies within a wide range, which can reflect the slight anisotropy hysteresis properties of the specimen. The effectiveness and accuracy of the improved vector model connected with the hybrid algorithm of PSO–Powell were verified.

By applying the hybrid algorithm, the slight error of the hysteresis loops simulated by the extracted parameters tended to increase at high *B*_m_ (more than 1 T), which was mainly attributed to the limitations of the proposed model. The model makes some improvements on the isotropic vector Preisach hysteresis model. It is noticed that the motion of the domain wall under the circular rotational excitation was more complicated with the increase of magnetic flux density (not only translation, but also rotation). The anisotropy properties of the material were more obvious, which makes accurate prediction by the proposed isotropic vector model difficult. Further improvements based on the classical vector hysteresis model, or application of the anisotropic vector model, are required for better accuracy.

## 4. Conclusions

The rotational magnetic characteristics of the SMC material are measured based on the experimental platform. Considering the anisotropic property of the material, an improved vector hysteresis model was put forward by introducing two parameters correlated with the amplitude of vector Everett function and the projection of magnetic flux density along different directions, to improve the shape of hysteresis curves.

Benefiting from the strong global search ability of the random optimization algorithm, in collaboration with fast local convergence of the deterministic optimization algorithm, a hybrid optimization strategy of PSO–Powell was proposed to extract the model parameters precisely and efficiently.

The simulated magnetic hysteresis characteristics of the SMC material under circular rotational magnetic excitation were basically consistent with the measured ones, which verified the effectiveness of the parameter extraction process by the hybrid optimization strategy and the improved model proposed in this paper.

## Figures and Tables

**Figure 1 materials-13-03138-f001:**
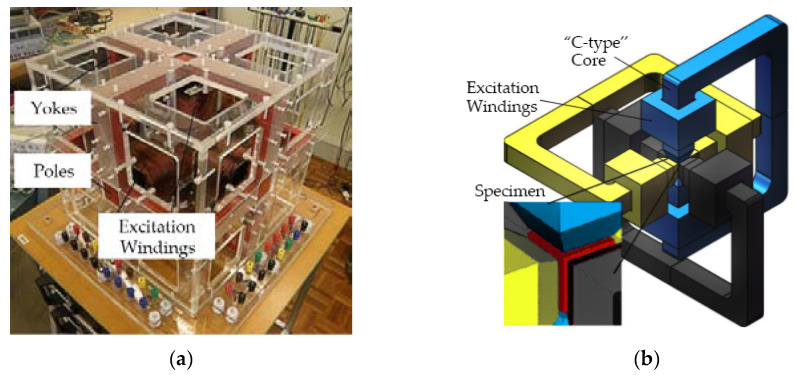
Experimental platform: (**a**) Main measuring system; (**b**) Diagrammatic sketch of magnetic circuit for measurement.

**Figure 2 materials-13-03138-f002:**
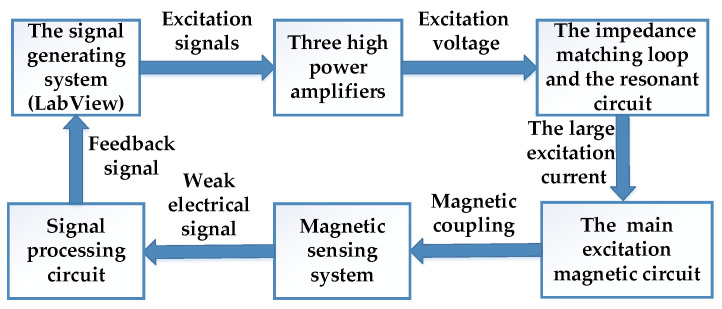
Structure block diagram of the measuring system of vector magnetic property.

**Figure 3 materials-13-03138-f003:**
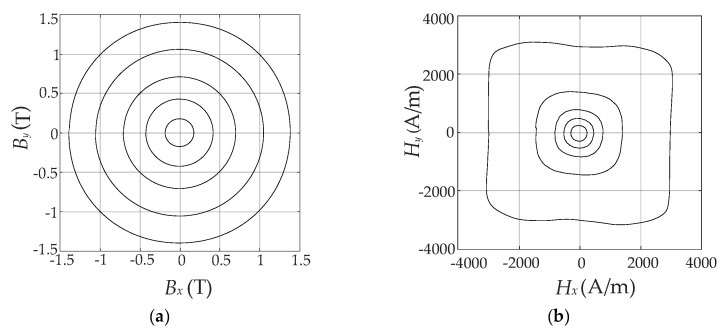
Measured hysteresis curves: (**a**) Locus of vector ***B***; (**b**) Locus of vector ***H***.

**Figure 4 materials-13-03138-f004:**
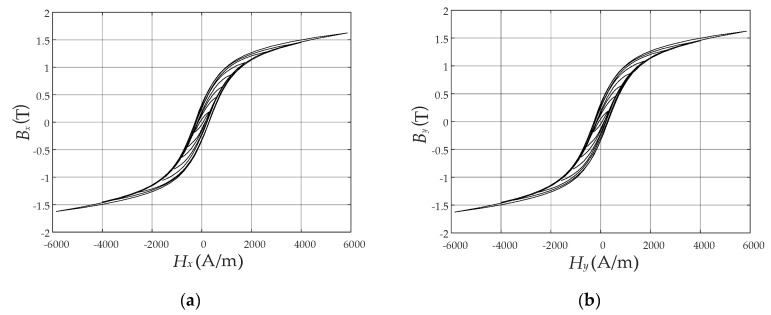
Measured concentric hysteresis loops in two orthogonal directions: (**a**) In *x* direction; (**b**) In *y* direction.

**Figure 5 materials-13-03138-f005:**
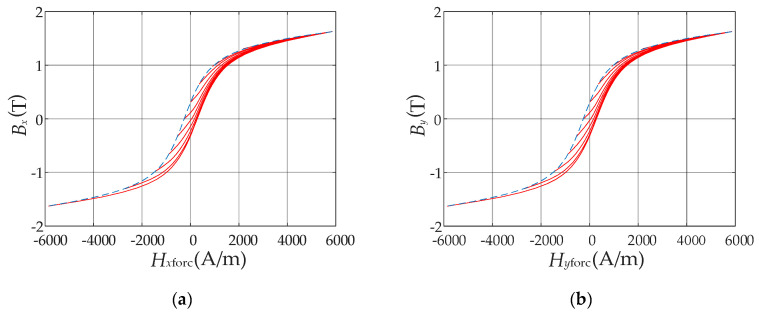
First-order reversal curves (FORCs) in two orthogonal directions: (**a**) In *x* direction; (**b**) In *y* direction.

**Figure 6 materials-13-03138-f006:**
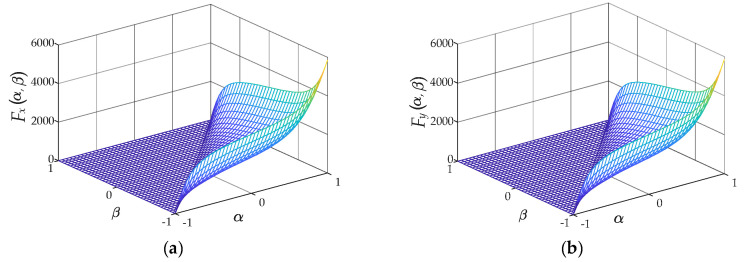
Normalized inverted scalar Everett function constructed from FORCs: (**a**) *F_x_*; (**b**) *F_y_*.

**Figure 7 materials-13-03138-f007:**
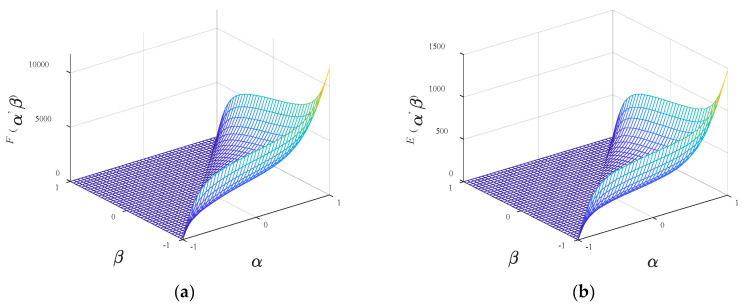
Normalized inverted scalar Everett function *F* and vector Everett function *E*: (**a**) Function *F*; (**b**) Function *E*.

**Figure 8 materials-13-03138-f008:**
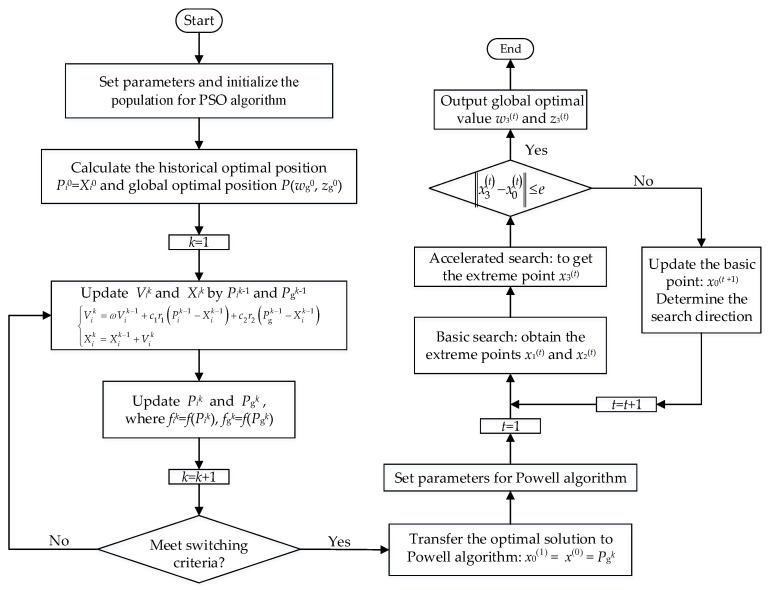
Flowchart of parameter extraction based on a hybrid algorithm of particle swarm optimization (PSO)–Powell.

**Figure 9 materials-13-03138-f009:**
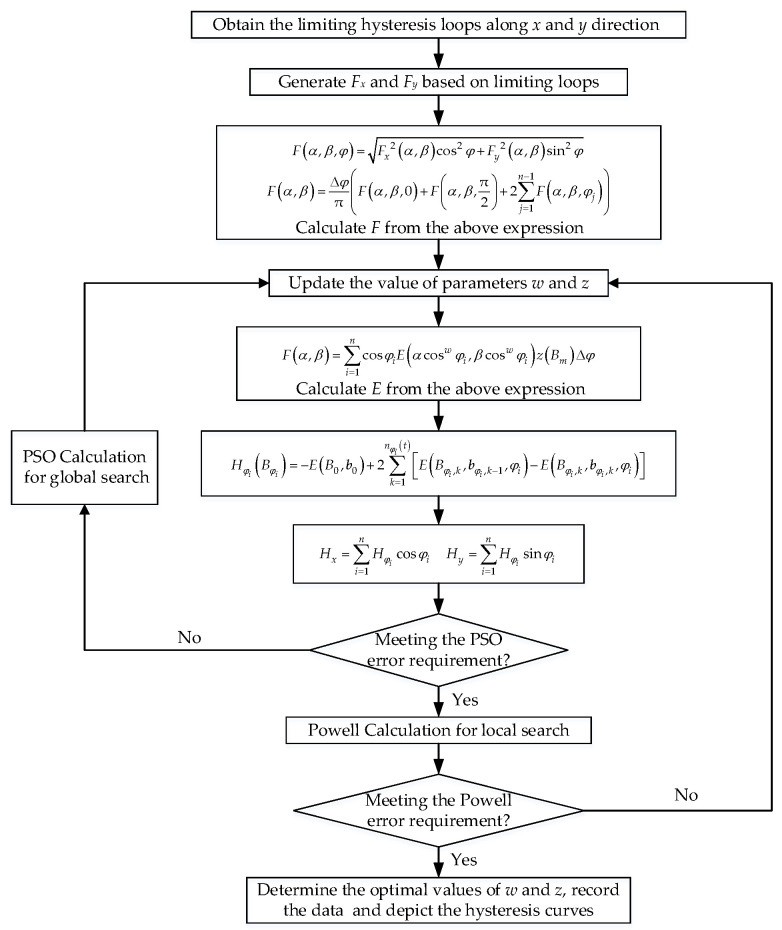
Flow diagram of simulation procedures.

**Figure 10 materials-13-03138-f010:**
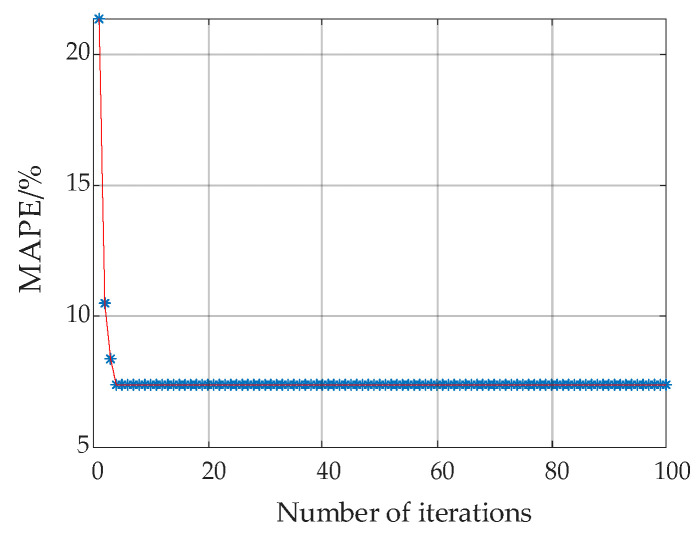
Variation of mean absolute percent error (MAPE) with number of iterations based on the PSO algorithm at 1.398T.

**Figure 11 materials-13-03138-f011:**
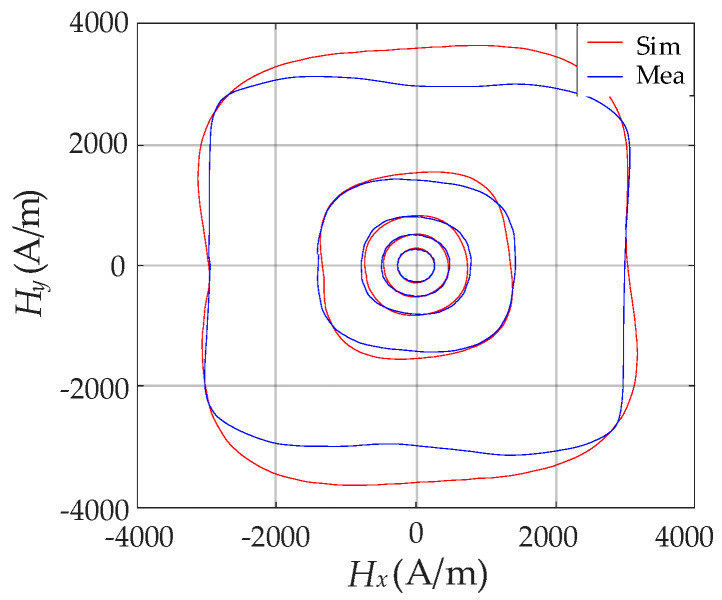
Comparison between measured ***H*** and simulated ***H*** based on the PSO algorithm.

**Figure 12 materials-13-03138-f012:**
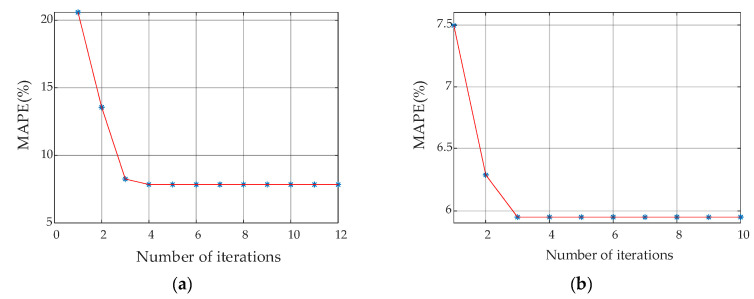
Variation of MAPE with iteration steps based on the PSO–Powell hybrid algorithm at 1.398 T: (**a**) Global optimization by PSO algorithm; (**b**) Local optimization by Powell algorithm.

**Figure 13 materials-13-03138-f013:**
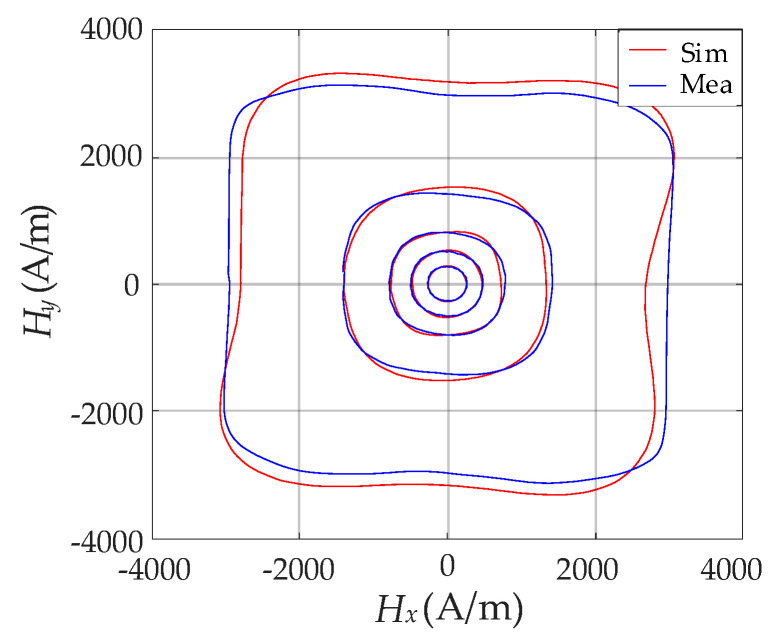
Comparison between experimental ***H*** and simulated ***H*** based on hybrid algorithm of PSO–Powell.

**Table 1 materials-13-03138-t001:** Values of parameters and corresponding error extracted by the PSO algorithm.

*B*_m_/T	*w*	*z*	MAPE/%	Bounds of Relative Error/%
1.398	0.7893	1.9567	7.3943	21.0209
1.053	0.7004	2.1219	4.9722	8.9200
0.704	0.6997	2.2231	4.0773	9.6547
0.423	0.5783	2.2117	3.3801	8.8839
0.178	0.6891	2.1583	2.8924	8.0391

**Table 2 materials-13-03138-t002:** Values of parameters and error extracted by hybrid algorithm of PSO–Powell.

*B*_m_/T	*w*	*z*	MAPE/%	Bounds of Relative Error/%
1.398	0.6951	2.1655	5.9471	10.7819
1.053	0.6960	2.1666	4.1672	8.2838
0.704	0.5755	2.1715	3.0289	7.0621
0.423	0.5939	2.1686	2.9272	6.3889
0.178	0.7182	2.1995	2.6848	6.5886

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
