# Peer review of "Anisotropic Vector Hysteresis Simulation of Soft Magnetic Composite Materials Based on a Hybrid Algorithm of PSO–Powell"

_materials, 2020, doi:10.3390/ma13143138_

Round 1

Reviewer 2 Report

Dear authors,

the manuscript is very well written. i would like to point out some minor things that coul be improved:

  • in some figures corresponding captions are on the next page
  • in fig. 1b) the zoom of the specimen could be improved by selecting more appropriate colors. i propose you leave the measurement yokes yellow, blue, black, and color the specimen red.
  • all the figures should have grids

Author Response

请参阅附件。

Reviewer 3 Report

The authors discuss results on vector hysteresis simulations of SMCs. Comments are given below:

  • Please check the material's name used because I believe it should read "SOMALLOY"
  • the data provided for the sample (coercivity, conductivity, etc.) is achievable at which compaction pressure/annealing temperature/time? Have you prepared the samples yourselves or it has been directly acquired from the supplier?
  • Please change "get rid" by eliminate;
  • after Equation (8): which numerical methods have been used?

Some general question: if a researcher has another sample shape and wishes to use the model described in the present paper, how s(he) proceed? I bring this point to your attention because it is quite common to have ring-shaped samples to be tested. Please comment on the manuscript.

Round 2

Reviewer 1 Report

I have checked your revised manuscript together with the letter of response. I have posed five major comments and three minor comments in my previous review. It seems to me that they are all sorted out and necessary information to understand the thoughts of the authors has been presented in the revised manuscript.

The English writing, however, should be improved significantly before it is accepted for publication by the editor. You started many sentences with a short connecting phrase in the revised manuscript. But I think excess use of short connecting phrase as a starter of sentences should better be avoided in the technical writing. Please try to include the connecting phrases inside a sentence to improve your writing to mean the same.

While I was checking your revised manuscript, I found the following small things that should be corrected. Please correct them if the editor accepts your manuscript for publication.

  • In page 1, 2nd paragraph of the introduction, "However, in practical", ... -> As "practical" is an adjective, it is strange. Do you want to mean "in practice" ?
  • Also in the 3rd paragraph of introduction, "However, the anisotropy magnetic properties under rotational excitations, is are found ... (as the subject "properties" are plural.)
  • In the same paragraph, "In addition, M. Enokizono and N. Soda ... considering the anisotropic under rotational magnetization." Is the usage of the word "anisotropic" correct in this sentence ? (anisotropic is an adjective)
  • According to Equation (6), The Everett function seemed to be a function of two variables alpha and beta. But three arguments (B, b, phi) are given to the function E in the revised version of Equation 7. I know what you want to mean with the three argument expression of E in Equation 7, but mathematical expression should be consistent throughout your manuscript.
